# Hospital volume and outcomes of surgical repair in type A acute aortic dissection: A nationwide cohort study

Iván Alejandro De León Ayala[1], Feng-Cheng Chang[2], Chun-Yu Chen[2], Yu-Ting Cheng[1], Yi-Hsin Chan[3], Victor Chien-Chia Wu[3], Kuo-Sheng Liu[1], Chi-Hsiao Yeh[1], Pao-Hsien Chu[3], Shao-Wei Chen[1,4]*

1 Division of Thoracic and Cardiovascular Surgery, Department of Surgery, Chang Gung Memorial Hospital, Linkou Medical Center, Chang Gung University, Taoyuan City, Taiwan, 2 Department of Anesthesiology, Chang Gung Memorial Hospital, Linkou Medical Center, Chang Gung University, Taoyuan City, Taiwan, 3 Department of Cardiology, Chang Gung Memorial Hospital, Linkou Medical Center, Chang Gung University, Taoyuan City, Taiwan, 4 Center for Big Data Analytics and Statistics, Chang Gung Memorial Hospital, Linkou Medical Center, Taoyuan City, Taiwan

* josephchen0314@gmail.com

## Abstract

### Background

Over the last decade, the number of patients treated with open repair for TAAAD in Taiwan has dramatically increased. This study aims to assess the hospital-volume relationship with surgical outcomes of type A acute aortic dissection (TAAAD) across hospitals in Taiwan.

### Methods

Using the Taiwan National Health Insurance (NHI) Research Database (NHIRD), we include only the patients who underwent first open repair for TAAAD from 01/01/2005, to 31/12/2020, in Taiwan. A total of 8,059 patients in 77 hospitals were eligible for the analysis.

Hospitals were categorized based on their 16-year cumulative volume of TAAAD open repair surgeries, and patients were grouped into quartiles accordingly.

### Results

Ascending aortic replacement (55.7%) and partial/total arch replacement (38.8%) were the most common methods of open aortic repair. In-hospital mortality was 22% and decreased from 28% in 2005 to 20% in 2020. Greater volume (per 5 annual surgeries) was associated with lower risks of in-hospital mortality (odd ratio 0.90, 95% confidence interval [CI] 0.87–0.92) and mortality after discharge (hazard ratio 0.97, 95% CI 0.95–0.99).

**Data availability statement:** The data underlying this study is from the NHIRD, which has been transferred to the HWDC. The NHIRD is not free to public access, and therefore interested researchers can obtain the data through formal application to the HWDC, Department of Statistics, Ministry of Health and Welfare, Taiwan (https://dep.mohw.gov.tw/DOS/cp-5119-59201-113.html). The authors had no special access privileges that others would not have.

**Funding:** This work was supported by grants from Chang Gung Memorial Hospital, Taiwan (CMRPG3P0801, CORPG3P0511, CORPG3P0541, CORPG3N0281, CORPG3N0282, CORPG3M0371, CORPG3M0372, CORPG3M0373, and BMRPD95 to SWC). Additional support was provided by the National Science and Technology Council, Taiwan (NSTC-112-2314-B-182A-107 and NSTC-113-2314-B-182A-087 to SWC), and by the Center for Big Data Analytics and Statistics, Chang Gung Memorial Hospital (CLRPG3N0011 to SWC). The funders had no role in study design, data collection and analysis, decision to publish, or preparation of the manuscript.

**Competing interests:** The authors have declared that no competing interests exist.

## Conclusion

Operative volume inversely correlates to in-hospital mortality and postoperative complications. The volume-outcome effect extends after discharge and reflects better long-term survival. Hospital referral to high-volume centers should be considered in patients needing complex open repair for TAAAD.

## Introduction

Acute type A dissection is a catastrophic condition requiring urgent treatment and immediate surgical intervention. Without surgery, medical treatment has nearly a 60% mortality within the first 24 hours after diagnosis [1]. Advances in diagnostic techniques and increased awareness have improved its detection and early treatment [2,3]. However, despite the efforts, postoperative mortality has been reported to be as high as 31% [4]. On the other hand, it has been reported that high-volume specialized centers can achieve postoperative results with in-hospital mortality below 10% [2,5–8].

The surgery for type A acute aortic dissection is complex, demanding knowledge of different surgical techniques (Like managing organ malperfusion) to achieve satisfactory outcomes. Moreover, the surgical learning curve to achieve a mortality of less than 10% is long, and this procedure requires continued exposure to maintain such outcomes [7,9–11]. Also, judicious planning with limited intervention based on surgical experience and the patient's clinical characteristics at presentation play a crucial role in achieving favorable surgical results.

Previous reports have addressed the effect of high volume and its impact on postoperative outcomes and in-hospital mortality [7,10–14]. However, there is less literature linking this high-volume experience to improved long-term survival. We present an analysis of a nationwide cohort across multiple Hospitals and throw-out time in Taiwan.

## Methods

### Ethical statement

The study was approved by the Institutional Review Board (IRB) of Chang Gung Memorial Hospital: IRB number 202201286B0, approved in 26/08/2022. The IRB committee waived the need for written patient informed consent.

### Data source

This retrospective cohort study used data from the Taiwan National Health Insurance (NHI) Research Database (NHIRD). The NHIRD includes data on all patients covered by the NHI program and is currently managed by the Health and Welfare Data Science Center (HWDC). More details about the NHIRD were previously described [15–18]. All personal information is deidentified and anonymized in the NHIRD, and informed consent was waived for participants. Following approval from IRB, data was accessed for analysis from 01/11/2022–01/12/2022. The data underlying this study is

from the NHIRD, which has been transferred to the HWDC. The NHIRD is not free to public access, and therefore interested researchers can obtain the data through formal application to the HWDC, Department of Statistics, Ministry of Health and Welfare, Taiwan (https://dep.mohw.gov.tw/DOS/cp-5119-59201-113.html). The authors had no special access privileges that others would not have.

## Study population

Patients admitted due to aortic dissection between 01/01/2005, and 31/12/2020, were included using the International Classification of Diseases, Clinical Modification, Ninth Revision (ICD-9-CM) and Tenth Revision (ICD-10-CM) diagnostic codes. We further ascertained patients who underwent open surgical repair for TAAAD using the Taiwan NHI reimbursements; a total of 9,642 patients were identified. Only patients undergoing open repair for TAAAD for the first time and with no history of previous aortic dissection were included. We did not include patients undergoing any other cardiac surgery activity. A total of 8,059 patients in 77 hospitals were eligible for the analysis. The cumulative hospital volume of TAAAD open repair surgeries was summed from 2005 to 2020, and the annual hospital volume was calculated. Patients were then evenly divided into four quartiles (The cutoffs were 10, 18.9 and 23.3 operations/year, respectively) based on the hospital where they were admitted and operated (Fig 1A).

## Outcomes

The primary outcome was in-hospital mortality, defined as death due to any reason during the index admission. The secondary outcomes included in-hospital complications and late outcomes. In-hospital complications included cardiogenic shock and the need for mechanical circulatory support (MCS), respiratory failure, new-onset stroke, new-onset dialysis, sepsis, deep wound infection, and massive blood transfusion (Defined as packed red blood cell > 10 U, [Single unit = 450–500cc]) during the index admission, length of hospital stay and intensive care unit (ICU), and medical expenditure. Late outcomes included all-cause mortality (including in-hospital mortality), all-cause mortality after surviving to discharge, aorta surgery (i.e., open repair and stent), all-cause readmission, and new-onset dialysis. The status, date, and causes of death were obtained via linking to the Taiwan Death Registry database in the HWDC. For the analysis of each late outcome, patients were followed until the day of death, event occurrence, or the end of the database (31/12/2020).

## Covariates

The covariates included demographics (i.e., age and sex), Marfan syndrome, comorbidities, events of history, year of surgery, the extension of aortic surgery (partial or total arch, elephant trunk, and aortic root replacement), and concomitant surgeries (i.e., coronary artery bypass graft and valve replacement [CABG]). Marfan syndrome was defined as having at least two outpatient diagnoses or any inpatient diagnosis during the entire database period (2000–2020). Comorbidities were defined as having at least two outpatient diagnoses or any inpatient diagnosis in the previous year, including hypertension, diabetes mellitus, atrial fibrillation, chronic kidney disease, and chronic obstructive pulmonary disease. History of events was defined as having any inpatient diagnosis before the index TAAAD admission, including heart failure, myocardial infarction, stroke, and cardiac surgery (CABG or valve).

## Statistical analysis

The trend of in-hospital mortality rates across the study period (2005–2020) was tested using Cochran-Armitage analysis. The relationship between average in-hospital mortality and annual hospital volume was assessed using Pearson's correlation. The patients were divided into four groups (quartiles) according to the annual hospital volume of open repair for TAAAD. The trend of patients' baseline characteristics and outcomes across the ordinal groups was tested using linear contrast of one-way analysis of variance for continuous variables or Cochran-Armitage analysis for categorical variables. The

A

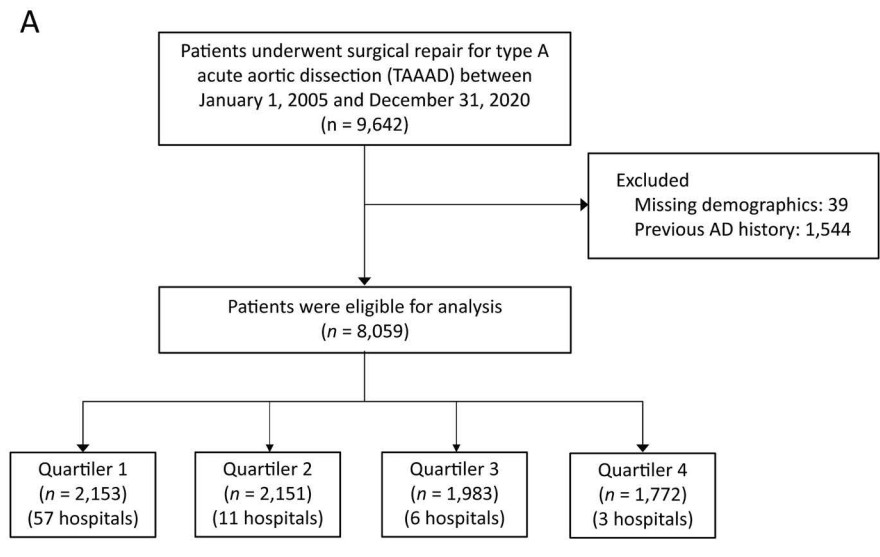

B

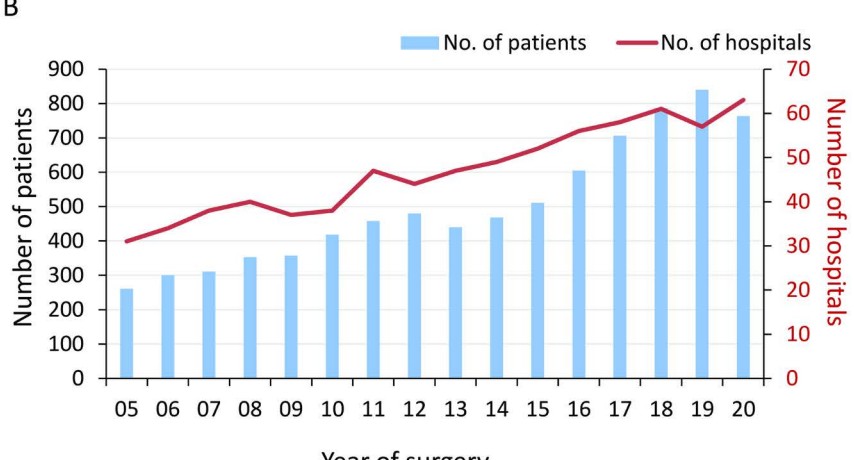

C

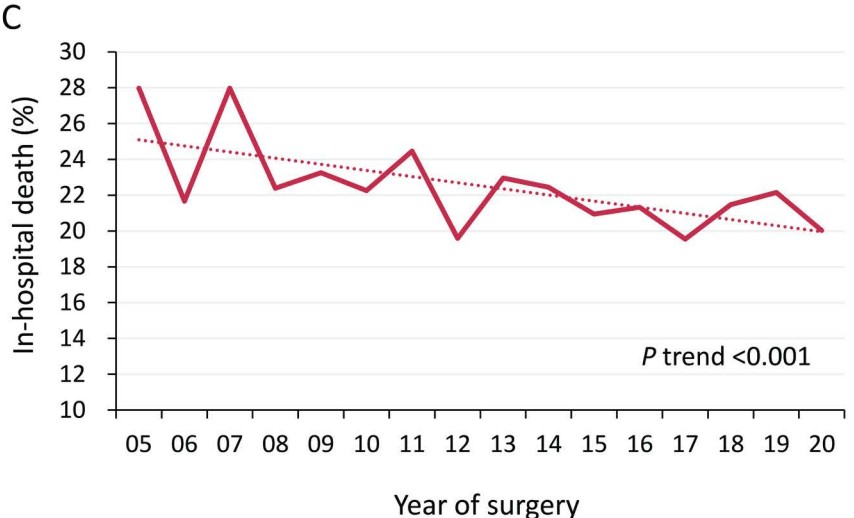

**Fig 1. Inclusion of study participants and surgical outcomes.** (A) Flowchart. (B) Number of patients and hospitals across time. (C) In-hospital mortality.

association between annual hospital volume (expressed as per 5 annual surgeries) and continuous outcomes (e.g., ICU duration), binary outcomes (e.g., in-hospital mortality), and late outcomes (e.g., redo aorta surgery) was evaluated using linear regression, logistic regression, and Cox proportional hazard model, respectively. All of the covariates were adjusted in the regression model mentioned above. We also compared patient outcomes between those who underwent surgery at the three major centers and those treated at hospitals with a very low surgical workload (fewer than 12 operations per year). We further conducted alternative Cox models, which treated annual hospital volume as a flexible restricted cubic spline (RCS) variable. The locations of knots were set at the 5th, 35th, 65th, and 95th percentiles. A two-sided P value <0.05 was considered to be statistically significant. Statistical analyses were performed using SAS version 9.4 (SAS Institute, Cary, NC). The RCS modeling was conducted using R version 4.0.2 (R Project for Statistical Computing) and the "rms" package.

## Results

### Open repair for TAAAD in Taiwan

During the study period (2005–2020), we included only patients receiving open repair for TAAAD and the hospitals performing open repair for TAAAD increased (Fig 1B). The number of hospitals was 31 in 2005 and 63 in 2020, respectively. The average in-hospital mortality rate significantly decreased from 28% in 2005 to 20% in 2020 (Fig 1C).

### Patient characteristics

Patients were younger in the hospitals with more volume (P trend <0.001), an extensive aortic repair was more frequent in hospitals with higher volume, particularly in the highest volume group (>23.3 annual surgeries, three hospitals) (Table 1).

### In-hospital outcomes

The results showed that more annual volume was significantly correlated to a lower average in-hospital mortality of the hospital (r = −0.32; Fig 2A). This phenomenon persisted in different eras, including 2005–2010, 2011–2015, and 2016–2020 (Fig 2B, 2C, and 2D). The overall in-hospital mortality was 22% (Table 2). The results showed that the greater volume (per 5 annual surgeries) was significantly associated with lower in-hospital mortality (Fig 3) (adjusted odd ratio 0.90, 95% confidence interval [CI] 0.87–0.92) after covariates adjustment. The RCS modeling revealed consistent findings (Fig 4A). Regarding other in-hospital outcomes, the greater volume was significantly associated with a lower risk of cardiogenic shock, new-onset dialysis, sepsis, massive transfusion, shorter ICU duration, longer hospitalization, and more in-hospital expenditure (P < 0.05).

### Late outcomes

The results showed that more annual volume was significantly associated with lower all-cause mortality (including in-hospital death) (hazard ratio [HR] 0.94, 95% CI 0.92–0.95) and all-cause mortality after surviving to discharge (HR 0.97, 95% CI 0.95–0.99) (Table 3). It is noticed that the all-cause mortality rate was slightly higher in the highest quartile than in the second-highest quartile (Fig 3A). However, the higher annual volume was significantly associated with a higher likelihood of redo aorta surgery, including open repair and endovascular stent. The RCS modeling demonstrated similar results (Fig 4B, 4C and 4D). In addition, there was no volume-outcome relationship between stroke, all-cause readmission, and new-onset dialysis during follow-up.

### Sensitivity analysis

After excluding hospitals with an annual surgical volume of fewer than five operations, the results demonstrated a significant inverse association between hospital surgical volume and average in-hospital mortality (Pearson's r = −0.42; S1 Fig). The results comparing patient outcomes between high-volume centers and low-volume hospitals are provided in

**Table 1. Baseline characteristics of the patients who underwent TAAAD surgery by the quartile of annual hospital volume.**

| Variable | Total | Q1 | Q2 | Q3 | Q4 | P trend |
|---|---|---|---|---|---|---|
| Range of annual hospital volume | – | ≤9.9 | 10-18.8 | 18.9-23.3 | >23.3 | – |
| Number of patients | 8059 | 2153 | 2151 | 1983 | 1772 | – |
| Number of hospitals | 77 | 57 | 11 | 6 | 3 | – |
| Age, year | 59.5±13.3 | 60.2±13.3 | 59.8±13.4 | 59.5±13.0 | 58.3±13.4 | <0.001 |
| Male | 5,435 (67.4) | 1,457 (67.7) | 1,443 (67.1) | 1,362 (68.7) | 1,173 (66.2) | 0.599 |
| Marfan syndrome | 141 (1.7) | 34 (1.6) | 28 (1.3) | 43 (2.2) | 36 (2.0) | 0.091 |
| Comorbidity | | | | | | |
| Hypertension | 6,052 (75.1) | 1,639 (76.1) | 1,596 (74.2) | 1,453 (73.3) | 1,364 (77.0) | 0.847 |
| Diabetes mellitus | 1,027 (12.7) | 280 (13.0) | 233 (10.8) | 218 (11.0) | 296 (16.7) | 0.003 |
| Atrial fibrillation | 538 (6.7) | 144 (6.7) | 161 (7.5) | 163 (8.2) | 70 (4.0) | 0.008 |
| Chronic kidney disease | 1,327 (16.5) | 392 (18.2) | 331 (15.4) | 310 (15.6) | 294 (16.6) | 0.177 |
| Chronic obstructive pulmonary disease | 446 (5.5) | 138 (6.4) | 126 (5.9) | 107 (5.4) | 75 (4.2) | 0.003 |
| Event of history | | | | | | |
| Heart failure | 359 (4.5) | 90 (4.2) | 95 (4.4) | 99 (5.0) | 75 (4.2) | 0.670 |
| Myocardial infarction | 153 (1.9) | 42 (2.0) | 41 (1.9) | 41 (2.1) | 29 (1.6) | 0.602 |
| Stroke | 561 (7.0) | 156 (7.2) | 155 (7.2) | 127 (6.4) | 123 (6.9) | 0.478 |
| Cardiac surgery | 55 (0.7) | 17 (0.8) | 14 (0.7) | 18 (0.9) | 6 (0.3) | 0.220 |
| Surgery year | | | | | | <0.001 |
| 2005-2007 | 872 (10.8) | 182 (8.5) | 221 (10.3) | 267 (13.5) | 202 (11.4) | |
| 2008-2010 | 1,128 (14.0) | 249 (11.6) | 297 (13.8) | 284 (14.3) | 298 (16.8) | |
| 2011-2013 | 1,378 (17.1) | 384 (17.8) | 371 (17.2) | 361 (18.2) | 262 (14.8) | |
| 2014-2016 | 1,584 (19.7) | 465 (21.6) | 405 (18.8) | 419 (21.1) | 295 (16.6) | |
| 2017-2020 | 3,097 (38.4) | 873 (40.5) | 857 (39.8) | 652 (32.9) | 715 (40.3) | |
| Extension of aortic surgery | | | | | | |
| Partial or total aortic arch replacement | 3,124 (38.8) | 771 (35.8) | 821 (38.2) | 677 (34.1) | 855 (48.3) | <0.001 |
| Elephant trunk (conventional or frozen) | 1,269 (15.7) | 261 (12.1) | 219 (10.2) | 281 (14.2) | 508 (28.7) | <0.001 |
| Aortic root replacement (Bentall operation) | 639 (7.9) | 172 (8.0) | 104 (4.8) | 146 (7.4) | 217 (12.2) | <0.001 |
| Ascending aorta replacement only | 4,488 (55.7) | 1,255 (58.3) | 1,256 (58.4) | 1,194 (60.2) | 783 (44.2) | <0.001 |
| Additional surgery | | | | | | |
| Coronary artery bypass graft | 754 (9.4) | 178 (8.3) | 208 (9.7) | 203 (10.2) | 165 (9.3) | 0.173 |
| Valve replacement | 1,098 (13.6) | 284 (13.2) | 248 (11.5) | 286 (14.4) | 280 (15.8) | 0.003 |
| Follow-up years | 4.2±4.2 | 3.5±4.0 | 4.0±4.1 | 4.9±4.3 | 4.4±4.3 | <0.001 |

Abbreviation: TAAAD, type A acute aortic dissection; Q, quartile.

Data were presented as mean±standard deviation or number and percentage.

the supplements (S1 and S2 Tables in S1 File). The results reinforced the primary analysis, indicating better mortality outcomes for patients who underwent surgery at major centers compared to those at low-volume hospitals. Among the top three hospitals with the highest-volume surgeries, the in-hospital mortality of the top first hospital was lower than the other two hospitals (Fig 2D); patients undergoing surgery in these two hospitals were excluded for further analysis. Baseline characteristics are provided in Table 4. The volume-outcome relationship was more evident after excluding these patients (Tables 5 and 6). Upon exclusion of the top two (or top three) highest-volume centers with unexpectedly high in-hospital mortality, the highest-volume quartile demonstrated a significantly lower all-cause mortality rate compared to the second-highest quartile (Fig 3B).

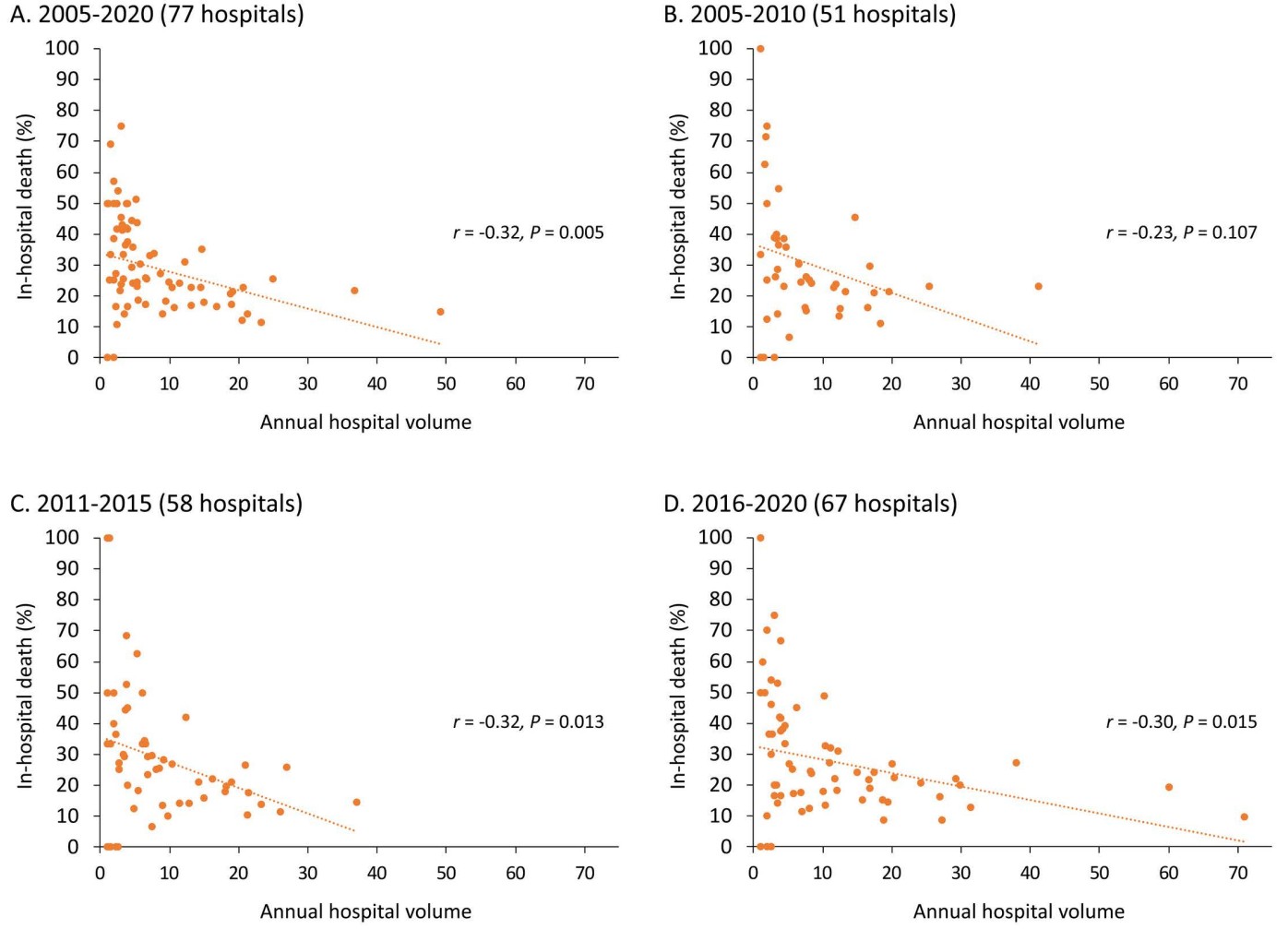

**Fig 2. Hospital annual volume and in-hospital mortality. (A)** Relationship between annual hospital volume and average in-hospital mortality. (B) 2005-2010. (C) 2011-2015. (D) 2016-2020.

## Discussion

It is well known that in high-volume centers, patients have a better immediate postoperative course and reduced in-hospital mortality after surgery for ATAAD [7,10–14]. However, it is not well understood to what extent a high-volume experience impacts the long-term survival of this group of patients. We present an analysis of a nationwide cohort across multiple Hospitals and throw-out time in Taiwan.

Over the last decade, the number of patients treated with open repair for TAAD in Taiwan has dramatically increased. From 2017 to 2020, the average of open repairs in Taiwan was 774 cases/year. Similarly to the USA [19], most of these surgeries in Taiwan are carried out within a small group of hospitals. However, only a few centers perform >20 surgeries/year, and less than five hospitals perform >50 cases/year.

The primary goal of the open repair of TAAD is to resect the entry tear to restore the flow through the true aortic lumen [20], with a limited approach to the elderly and more critical patient [6,21,22] or as suggested: performing aortic root replacement only if irreparable aortic root, a root diameter ≥4.5 cm with or without connective tissue disease, and an unrepairable aortic valve [8,23], reserving more aggressive and complex procedures (Bentall procedure, total arch

**Table 2. In-hospital outcomes of the patients who underwent TAAAD surgery by the quartile of annual hospital volume.**

| Variable | Total | Q1 | Q2 | Q3 | Q4 | P trend | Adjusted OR/B (95% CI)* | P value |
|---|---|---|---|---|---|---|---|---|
| Number of patients | 8,059 | 2,153 | 2,151 | 1,983 | 1,772 | | | |
| Primary outcome | | | | | | | | |
| In-hospital mortality | 1,774 (22.0) | 631 (29.3) | 472 (21.9) | 324 (16.3) | 347 (19.6) | <0.001 | 0.90 (0.87–0.92) | <0.001 |
| Secondary outcome | | | | | | | | |
| Cardiogenic shock and need MCS | 841 (10.4) | 285 (13.2) | 177 (8.2) | 157 (7.9) | 222 (12.5) | 0.233 | 0.96 (0.93–0.99) | 0.004 |
| Respiratory failure | 1,464 (18.2) | 380 (17.6) | 332 (15.4) | 204 (10.3) | 548 (30.9) | <0.001 | 1.01 (0.99–1.03) | 0.516 |
| New onset stroke | 940 (11.7) | 244 (11.3) | 260 (12.1) | 200 (10.1) | 236 (13.3) | 0.279 | 1.04 (1.01–1.06) | 0.010 |
| Dialysis (de novo dialysis) | 1,392 (17.3) | 380 (17.6) | 367 (17.1) | 327 (16.5) | 318 (17.9) | 1.000 | 0.97 (0.94–0.99) | 0.008 |
| Sepsis | 422 (5.2) | 125 (5.8) | 129 (6.0) | 96 (4.8) | 72 (4.1) | 0.005 | 0.95 (0.91–0.99) | 0.012 |
| Deep wound infection | 215 (2.7) | 43 (2.0) | 72 (3.3) | 52 (2.6) | 48 (2.7) | 0.348 | 1.02 (0.97–1.08) | 0.363 |
| Massive transfusion (PRBC ≥ 10U) | 2,848 (35.3) | 829 (38.5) | 734 (34.1) | 550 (27.7) | 735 (41.5) | 0.9772 | 0.96 (0.95–0.98) | <0.001 |
| ICU duration (days) | 10.9±14.8 | 10.4±13.9 | 11.6±13.3 | 11.1±18.3 | 10.3±13.1 | 0.750 | −0.19 (−0.32, −0.07) | 0.002 |
| In-hospital stay, day | 27.9±32.2 | 24.8±35.5 | 28.6±28.4 | 29.3±34.3 | 29.1±29.5 | <0.001 | 0.45 (0.18, 0.72) | 0.001 |
| In-hospital cost (NTD × 10⁴) | 24.2±17.0 | 21.9±14.0 | 23.2±13.5 | 24.4±19.4 | 28.1±20.1 | <0.001 | 0.36 (0.22, 0.50) | <0.001 |

Abbreviation: TAAAD, type A acute aortic dissection; Q, quartile; OR, odds ratio; B, regression coefficient; CI, confidence interval; MCS, mechanical circulatory support; PRBC, packed red blood cells; ICU, intensive care unit; NTD, New Taiwan Dollar.

Data were presented as mean±standard deviation or number and percentage.

*Associated with per 5 annual surgeries increased with adjustment of all covariates listed in **Table 1**.

replacement, or frozen elephant trunk) to the young and more stable patient [6]. Our report found this evident when analyzing high-volume centers routinely performing complex procedures versus high-volume centers adopting a more tailored approach.

The open repair of TAAAD is a very demanding surgery that requires continued exposure to achieve and maintain satisfactory results [9]. It is well known that hospital volume correlates with patient outcomes [7,10,12,13] and that there is an inverse relationship between hospital volume and in-hospital mortality [10,11,13]. Even further, the outcomes of high-volume and low-volume surgeons are not the same, even in a high-volume center environment [7,10].

When a surgeon performs >5 surgeries/year, the postoperative mortality falls below 18% [7]. Thus, it seems that the critical point is not only high-volume centers but high-volume centers with a high volume of aortic surgery performed by high-volume aortic surgeons. This might suggest that a specialized team of aortic dissection should be implemented in high-volume centers to improve the outcomes further. Also, besides the importance of surgical experience, high-volume hospitals might have the advantage of better postoperative care due to differences in practice patterns in and out of the operating room [11], established protocols, and specialized staff, lowering further in-hospital mortality and complications.

Reducing the time between onset and surgery is especially important in TAAD [2]. However, evaluating the need for medical referral to high-volume aortic centers is essential when facing critical patients needing a more complex surgical repair. There is an absolute risk reduction in operative mortality in high-volume centers, with a continued benefit during long-term follow-up. However, in Taiwan, there is no regionalization policy dictating the referral of patients to high-volume hospitals instead of low-volume aortic surgery centers. Thus, despite the obvious short- and long-term benefits of referring patients to high-volume centers [24], Taiwan's health system does not restrict where patients can undergo surgery for acute aortic dissection. It would be worthwhile for Taiwan's health authorities to consider pilot programs or phased approaches to regionalize the surgical treatment of acute type A aortic dissection.

Like the experience in Japan [5], surgical mortality in Taiwan decreased over time. In our report, regardless of the era, high-volume centers persist in achieving lower in-hospital mortality and better outcomes than low-volume centers.2

## A. The entire cohort (all 77 hospitals)

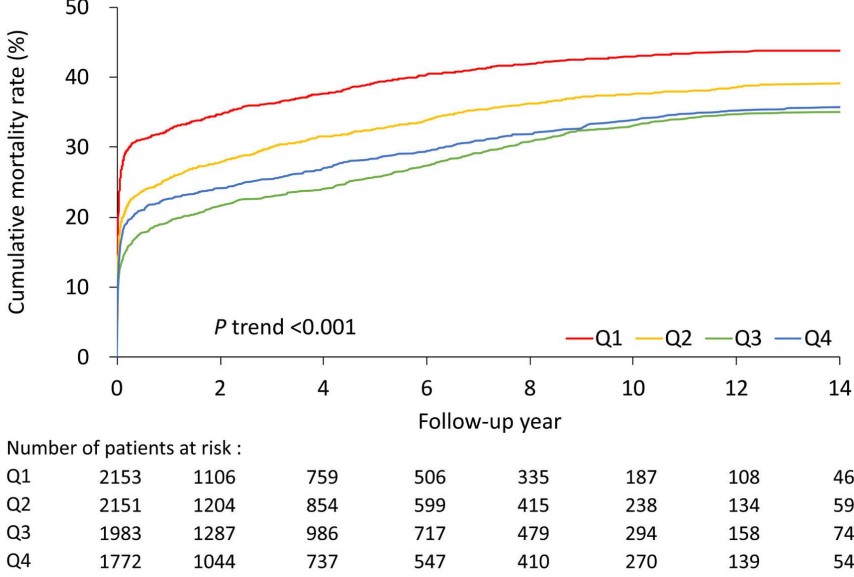

| Number of patients at risk : | | | | | | | | |
|---|---|---|---|---|---|---|---|---|
| Q1 | 2153 | 1106 | 759 | 506 | 335 | 187 | 108 | 46 |
| Q2 | 2151 | 1204 | 854 | 599 | 415 | 238 | 134 | 59 |
| Q3 | 1983 | 1287 | 986 | 717 | 479 | 294 | 158 | 74 |
| Q4 | 1772 | 1044 | 737 | 547 | 410 | 270 | 139 | 54 |

## B. Excluding the 2nd and 3rd most (75 hospitals)

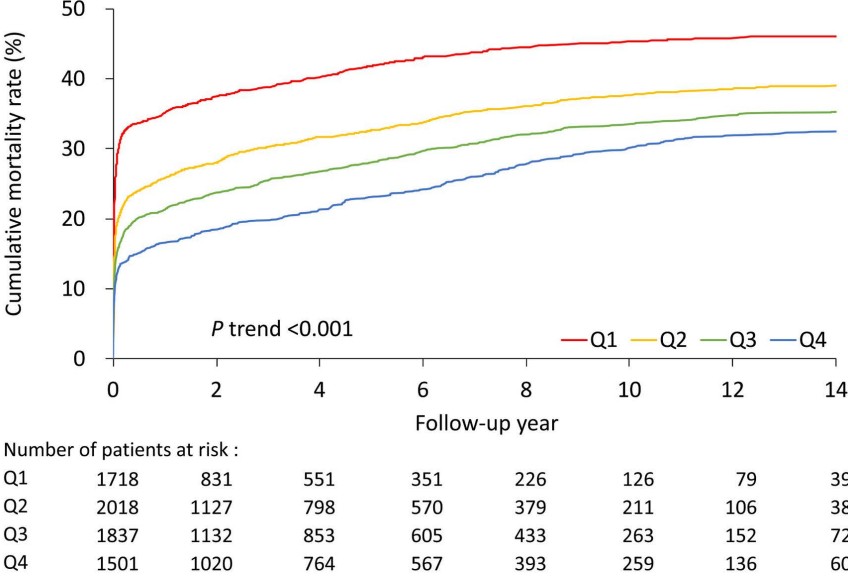

| Number of patients at risk : | | | | | | | | |
|---|---|---|---|---|---|---|---|---|
| Q1 | 1718 | 831 | 551 | 351 | 226 | 126 | 79 | 39 |
| Q2 | 2018 | 1127 | 798 | 570 | 379 | 211 | 106 | 38 |
| Q3 | 1837 | 1132 | 853 | 605 | 433 | 263 | 152 | 72 |
| Q4 | 1501 | 1020 | 764 | 567 | 393 | 259 | 136 | 60 |

**Fig 3. Mortality by quartiles.** (A) Cumulative mortality of the whole cohort. (B) Exclusion of the top two and top three hospitals. Q, quartile.

Interestingly, the new onset of stroke was higher in the high-volume group; this could be explained by the higher number of image studies (Brain CT or MRI) performed at high-volume hospitals.

When analyzing the data, one should consider the learning curve of the open repair of TAAA.

Our previous study demonstrated [9] that for a surgeon to achieve acceptable outcomes (Mortality <10%), a surgeon should perform at least 25 surgeries; moreover, decreased exposure to this surgery correlates with increased mortality, even after completing the training period.

A. In-hospital death

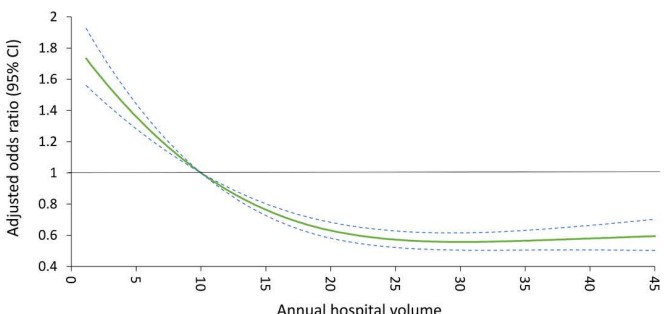

B. All-cause mortality

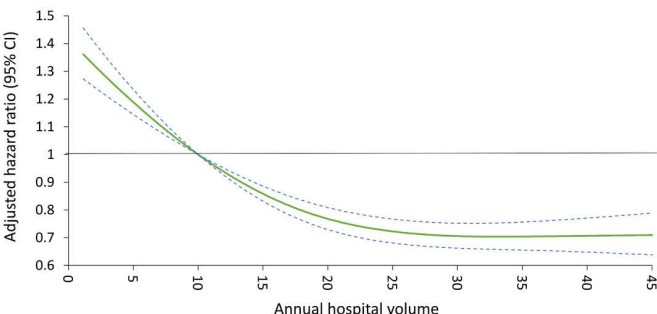

D. Any aorta surgery (open or stent)

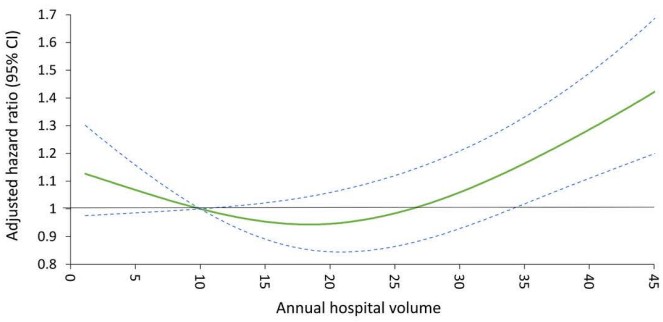

C. All-cause mortality after surviving to discharge

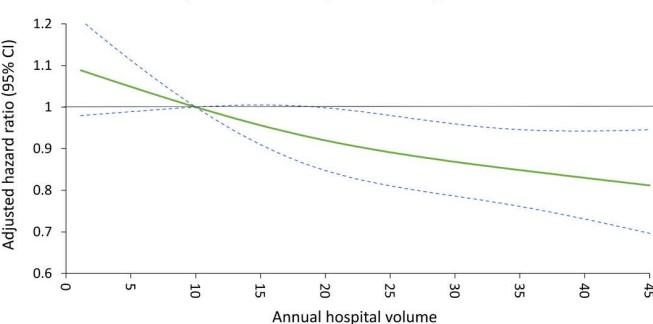

**Fig 4. Overall mortality and reoperation.** (A) The relationship between annual hospital volume and the risk of in-hospital death. (B) all-cause mortality, including in-hospital death. (C) All-cause mortality after surviving to discharge. (D) Redo aorta surgery (Open repair or endovascular stent). CI, confidence interval.

**Table 3. Late outcomes of the patients who underwent TAAAD surgery by the quartile of annual hospital volume.**

| Variable | Total | Q1 | Q2 | Q3 | Q4 | P trend | Adjusted HR (95% CI)* | P value |
|---|---|---|---|---|---|---|---|---|
| Number of patients | 8,059 | 2,153 | 2,151 | 1,983 | 1,772 | | | |
| All-cause mortality | 3,132 (38.9) | 951 (44.2) | 845 (39.3) | 702 (35.4) | 634 (35.8) | <0.001 | 0.94 (0.92–0.95) | <0.001 |
| Mortality after discharge | 1,426 (17.7) | 345 (16.0) | 392 (18.2) | 387 (19.5) | 302 (17.0) | 0.010 | 0.97 (0.95–0.99) | 0.002 |
| Stroke | 477 (5.9) | 108 (5.0) | 144 (6.7) | 110 (5.5) | 115 (6.5) | 0.595 | 1.003 (0.97–1.04) | 0.856 |
| Aorta surgery | | | | | | | | |
| Any aorta surgery | 832 (10.3) | 189 (8.8) | 167 (7.8) | 238 (12.0) | 238 (13.4) | <0.001 | 1.05 (1.02–1.08) | <0.001 |
| Open repair | 233 (2.9) | 45 (2.1) | 47 (2.2) | 77 (3.9) | 64 (3.6) | 0.012 | 1.06 (1.01–1.11) | 0.025 |
| Stent | 653 (8.1) | 156 (7.2) | 130 (6.0) | 175 (8.8) | 192 (10.8) | 0.009 | 1.04 (1.01–1.07) | 0.008 |
| Readmission due to any cause | 3,975 (49.3) | 947 (44.0) | 1,050 (48.8) | 1,062 (53.6) | 916 (51.7) | 0.334 | 0.997 (0.99–1.01) | 0.623 |
| New-onset dialysis | 121 (1.5) | 22 (1.0) | 38 (1.8) | 33 (1.7) | 28 (1.6) | 0.830 | 1.01 (0.95–1.09) | 0.706 |

Abbreviation: TAAAD, type A acute aortic dissection; Q, quartile; HR, hazard ratio; CI, confidence interval.

Data were presented as numbers and percentages.

*Associated with per 5 annual surgeries increased with adjustment of all covariates listed in **Table 1**.

**Table 4. Baseline characteristics of the patients who underwent TAAAD surgery by the quartile of annual hospital volume, excluding the second and third most volume hospitals.**

| Variable | Total | Q1 | Q2 | Q3 | Q4 | *P* trend |
|---|---|---|---|---|---|---|
| Range of annual hospital volume | – | ≤8.9 | 9.0-14.9 | 15.0-20.7 | >20.7 | – |
| Number of patients | 7074 | 1718 | 2018 | 1837 | 1501 | – |
| Number of hospitals | 75 | 54 | 12 | 6 | 3 | – |
| Age, years | 59.5±13.3 | 60.1±13.2 | 60.3±13.4 | 59.1±12.9 | 58.3±13.7 | <.001 |
| Male | 4,790 (67.7) | 1,164 (67.8) | 1,343 (66.6) | 1,264 (68.8) | 1,019 (67.9) | .56 |
| Comorbidity | | | | | | |
| Marfan syndrome | 135 (1.9) | 27 (1.6) | 24 (1.2) | 31 (1.7) | 53 (3.5) | <.001 |
| Hypertension | 5,263 (74.4) | 1,294 (75.3) | 1,526 (75.6) | 1,361 (74.1) | 1,082 (72.1) | .021 |
| Diabetes mellitus | 811 (11.5) | 232 (13.5) | 222 (11.0) | 197 (10.7) | 160 (10.7) | .012 |
| Heart failure | 316 (4.5) | 75 (4.4) | 86 (4.3) | 87 (4.7) | 68 (4.5) | .64 |
| Old myocardial infarction | 134 (1.9) | 37 (2.2) | 40 (2.0) | 25 (1.4) | 32 (2.1) | .56 |
| Atrial fibrillation | 502 (7.1) | 116 (6.8) | 135 (6.7) | 157 (8.5) | 94 (6.3) | .74 |
| Old stroke | 494 (7.0) | 137 (8.0) | 137 (6.8) | 126 (6.9) | 94 (6.3) | .076 |
| Chronic kidney disease | 1,157 (16.4) | 327 (19.0) | 321 (15.9) | 290 (15.8) | 219 (14.6) | .001 |
| Chronic obstructive pulmonary disease | 398 (5.6) | 104 (6.1) | 123 (6.1) | 108 (5.9) | 63 (4.2) | .029 |
| Previous cardiac surgery | 52 (0.7) | 17 (1.0) | 12 (0.6) | 11 (0.6) | 12 (0.8) | .52 |
| Surgery year | | | | | | <.001 |
| 2005-2007 | 783 (11.1) | 146 (8.5) | 174 (8.6) | 235 (12.8) | 228 (15.2) | |
| 2008-2010 | 964 (13.6) | 167 (9.7) | 283 (14.0) | 268 (14.6) | 246 (16.4) | |
| 2011-2013 | 1,228 (17.4) | 289 (16.8) | 366 (18.1) | 325 (17.7) | 248 (16.5) | |
| 2014-2016 | 1,433 (20.3) | 381 (22.2) | 393 (19.5) | 369 (20.1) | 290 (19.3) | |
| 2017-2020 | 2,666 (37.7) | 735 (42.8) | 802 (39.7) | 640 (34.8) | 489 (32.6) | |
| Type A dissection surgical detail | | | | | | |
| Extension of aortic surgery | | | | | | |
| Partial or total aortic arch replacement | 2,513 (35.5) | 592 (34.5) | 729 (36.1) | 783 (42.6) | 409 (27.2) | .021 |
| Elephant trunk (conventional or frozen) | 794 (11.2) | 180 (10.5) | 250 (12.4) | 249 (13.6) | 115 (7.7) | .073 |
| Aortic root replacement (Bentall operation) | 496 (7.0) | 141 (8.2) | 121 (6.0) | 115 (6.3) | 119 (7.9) | .76 |
| Ascending aorta replacement only | 4,187 (59.2) | 1,017 (59.2) | 1,210 (60.0) | 967 (52.6) | 993 (66.2) | .046 |
| Additional surgery | | | | | | |
| Coronary artery bypass graft | 642 (9.1) | 145 (8.4) | 180 (8.9) | 197 (10.7) | 120 (8.0) | .73 |
| Valve replacement | 912 (12.9) | 225 (13.1) | 240 (11.9) | 271 (14.8) | 176 (11.7) | .88 |
| Follow-up years | 4.2±4.2 | 3.2±3.8 | 4.0±4.0 | 4.6±4.4 | 5.1±4.4 | <.001 |

Abbreviation: TAAAD, type A acute aortic dissection; Q, quartile.

Data were presented as mean±standard deviation or number and percentage.

Aortic surgery reinterventions are much more complex, with an increased risk of surgical complications and postoperative mortality. In our data, high-volume centers had an increased reoperation rate; this could be explained by the fact that high-volume centers have more experience, better follow-up protocols, and might be more active in performing reoperations, as previously suggested [12].

Limitations: First, due to inherent limitations of using the Taiwan National Health Insurance (NHI) Research Database (NHIRD), we did not had access to information regarding clinical status at presentation (shock, neurology status, etc..). Disease severity was not available for study, and we did not have access to the preoperative CT scans in our study, thus, limiting stratification in our study. Second, this study has the limitations inherited from its retrospective design; however,

**Table 5. In hospital outcomes of the patients who underwent TAAAD surgery by the quartile of annual hospital volume, excluding the second and third most volume hospitals.**

| Variable | Total | Q1 | Q2 | Q3 | Q4 | P trend | Adjusted OR/B (95% CI)* | P value |
|---|---|---|---|---|---|---|---|---|
| Number of patients | 7074 | 1718 | 2018 | 1837 | 1501 | | | |
| Categorical parameter | | | | | | | | |
| In hospital mortality | 1,545 (21.8) | 547 (31.8) | 450 (22.3) | 339 (18.5) | 209 (13.9) | <0.001 | 0.87 (0.85–0.90) | <.001 |
| Cardiogenic shock and needed MCS | 671 (9.5) | 232 (13.5) | 188 (9.3) | 165 (9.0) | 86 (5.7) | <0.001 | 0.90 (0.86–0.93) | <.001 |
| Respiratory failure | 964 (13.6) | 298 (17.3) | 298 (14.8) | 278 (15.1) | 90 (6.0) | <0.001 | 0.86 (0.83–0.89) | <.001 |
| New onset stroke | 829 (11.7) | 196 (11.4) | 222 (11.0) | 227 (12.4) | 184 (12.3) | 0.253 | 1.05 (1.02–1.08) | <.001 |
| Dialysis (de novo dialysis) | 1,162 (16.4) | 314 (18.3) | 382 (18.9) | 289 (15.7) | 177 (11.8) | <0.001 | 0.93 (0.91–0.96) | <.001 |
| Sepsis | 385 (5.4) | 95 (5.5) | 131 (6.5) | 111 (6.0) | 48 (3.2) | 0.005 | 0.96 (0.91–0.997) | .037 |
| Deep wound infection | 196 (2.8) | 28 (1.6) | 70 (3.5) | 49 (2.7) | 49 (3.3) | 0.029 | 1.05 (0.995–1.11) | .077 |
| Massive transfusion (PRBC ≥ 10U) | 2,311 (32.7) | 697 (40.6) | 741 (36.7) | 532 (29.0) | 341 (22.7) | <0.001 | 0.91 (0.89–0.93) | <.001 |
| Continuous parameter | | | | | | | | |
| ICU duration (days) | 10.7±15.0 | 10.2±13.4 | 11.9±14.4 | 11.7±18.0 | 8.7±12.9 | 0.010 | −0.21 (−0.34, −0.07) | .003 |
| In-hospital stay | 27.6±32.6 | 24.0±37.3 | 29.1±29.2 | 29.4±32.4 | 27.7±31.0 | 0.001 | 0.57 (0.27, 0.86) | <.001 |
| In-hospital cost (NTD × $10^4$) | 23.0±15.7 | 21.6±14.0 | 23.7±14.0 | 24.7±18.9 | 21.5±14.8 | 0.669 | 0.14 (−0.01, 0.28) | .063 |

Abbreviation: TAAAD, type A acute aortic dissection; Q, quartile; OR, odds ratio; B, regression coefficient; CI, confidence interval; MCS, mechanical circulatory support; PRBC, packed red blood cells; ICU, intensive care unit; NTD, New Taiwan Dollar;

Data were presented as mean±standard deviation or number and percentage.

*Associated with per 5 annual surgeries increased with adjustment of all covariates listed in **Table 3**.

**Table 6. Late outcomes of the patients who underwent TAAAD surgery by the quartile of annual hospital volume, excluding the second and third most volume hospitals.**

| Variable | Total | Q1 | Q2 | Q3 | Q4 | P trend | Adjusted HR (95% CI)* | P value |
|---|---|---|---|---|---|---|---|---|
| Number of patients | 7074 | 1718 | 2018 | 1837 | 1501 | | | |
| All-cause mortality | 2,732 (38.6) | 798 (46.4) | 790 (39.1) | 652 (35.5) | 492 (32.8) | <0.001 | 0.92 (0.91–0.94) | <.001 |
| Mortality after discharge | 1,245 (17.6) | 273 (15.9) | 355 (17.6) | 326 (17.7) | 291 (19.4) | <0.001 | 0.96 (0.94–0.98) | <.001 |
| Stroke | 417 (5.9) | 84 (4.9) | 126 (6.2) | 114 (6.2) | 93 (6.2) | 0.100 | 0.996 (0.96–1.04) | .85 |
| Aorta surgery | | | | | | | | |
| Any aorta surgery | 218 (3.1) | 35 (2.0) | 47 (2.3) | 55 (3.0) | 81 (5.4) | 0.001 | 1.08 (1.03–1.13) | .002 |
| Open repair | 542 (7.7) | 131 (7.6) | 115 (5.7) | 166 (9.0) | 130 (8.7) | 0.159 | 1.01 (0.98–1.05) | .48 |
| Stent | 713 (10.1) | 155 (9.0) | 154 (7.6) | 209 (11.4) | 195 (13.0) | 0.497 | 1.04 (1.01–1.07) | .012 |
| Readmission due to any cause | 3,482 (49.2) | 715 (41.6) | 986 (48.9) | 961 (52.3) | 820 (54.6) | 0.002 | 0.992 (0.98–1.01) | .22 |
| New-onset dialysis | 105 (1.5) | 19 (1.1) | 26 (1.3) | 36 (2.0) | 24 (1.6) | 0.989 | 1.01 (0.94–1.09) | .770 |

Abbreviation: TAAAD, type A acute aortic dissection; Q, quartile; HR, hazard ratio; CI, confidence interval.

Data were presented as numbers and percentages.

*Associated with per 5 annual surgeries increased with adjustment of all covariates listed in **Table 3**

this is a nationwide cohort across hospitals and throw-out time in Taiwan. Third, we did not have data on individual surgeons' volume, and the surgeon's profile was not adjusted for the comparison. Fourth, this report faced restrictions linked to the health insurance system in Taiwan, its way of classifying diseases, and its referral system. Our results might not reflect the conditions in other countries under different health systems.

## Conclusions

Operative volume inversely correlates to in-hospital mortality and postoperative complications, and this effect remained long after discharge and during follow-up. Hospital referral to high-volume centers should be considered in patients needing complex open repair for TAAAD. An aortic dissection specialized training/subspeciality should be encouraged in high-volume centers.

## Supporting information

**S1 Fig.  Sensitivity analysis.** Hospitals with an annual surgical volume of five or more operations.
(TIF)

**S1 File.   Supplemental Table 1**. In-hospital outcomes  of the patients  between those who underwent surgery at the three major centers and those treated at hospitals with a very low surgical workload (fewer than 12 operations per year). **Supplemental Table 2**. Late outcomes of the patients  of the patients between those who underwent surgery at the three major centers and those treated at hospitals with a very low surgical workload (fewer than 12 operations per year).
(DOCX)

## Acknowledgments

This study was based on data from the NHIRD provided by the NHI administration, Ministry of Health and Welfare of Taiwan. However, the interpretation and conclusions contained in this paper only represent the authors. We would like to thank the Maintenance Project of the Center for Big Data Analytics and Statistics at Chang Gung Memorial Hospital and Alfred Hsing-Fen Lin and Jia-Kai Wu for their assistance with the statistical analysis.

## Author contributions

**Conceptualization:** Shao-Wei Chen.

**Data curation:** Yu-Ting Cheng, Victor Chien-Chia Wu, Shao-Wei Chen.

**Formal analysis:** Shao-Wei Chen.

**Funding acquisition:** Feng-Cheng Chang, Shao-Wei Chen.

**Investigation:** Iván Alejandro De León Ayala, Feng-Cheng Chang, Shao-Wei Chen.

**Methodology:** Iván Alejandro De León Ayala, Feng-Cheng Chang.

**Project administration:** Shao-Wei Chen.

**Resources:** Chun-Yu Chen, Yi-Hsin Chan, Kuo-Sheng Liu, Chi-Hsiao Yeh, Pao-Hsien Chu.

**Supervision:** Chun-Yu Chen, Yi-Hsin Chan, Kuo-Sheng Liu, Chi-Hsiao Yeh, Pao-Hsien Chu.

**Validation:** Iván Alejandro De León Ayala, Feng-Cheng Chang.

**Writing – original draft:** Iván Alejandro De León Ayala.

**Writing – review & editing:** Iván Alejandro De León Ayala, Feng-Cheng Chang, Shao-Wei Chen.

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
