## [Decision Letter · Decision Letter 0]

19 Feb 2025

Dear Dr. Chen,

Thank you for submitting your manuscript to PLOS ONE. After careful consideration, we feel that it has merit but does not fully meet PLOS ONE’s publication criteria as it currently stands. Therefore, we invite you to submit a revised version of the manuscript that addresses the points raised during the review process.

We look forward to receiving your revised manuscript.

Kind regards,

Redoy Ranjan, MBBS, MRCSEd, Ch.M., MS (CV&TS), FACS

Academic Editor

PLOS ONE

Journal Requirements:

3. Thank you for stating the following financial disclosure: [This work was supported by a grant from Chang Gung Memorial Hospital, Taiwan CORPG3P0521(FCC) and CMRPG3P0801, CORPG3P0511, CORPG3P0541, CORPG3N0281, CORPG3N0282, CORPG3M0371, CORPG3M0372, CORPG3M0373, BMRPD95(SWC). This work was also supported by National Science and Technology Council grant NSTC-113-2314-B-182A-017 (FCC) and NSTC-112-2314-B-182A-107, NSTC-113-2314-B-182A-087 (SWC)].

4. Thank you for stating the following in the Acknowledgments Section of your manuscript: [This study was based on data from the NHIRD provided by the NHI administration, Ministry of Health and Welfare of Taiwan. However, the interpretation and conclusions contained in this paper only represent the authors. We would like to thank the Maintenance Project of the Center for Big Data Analytics and Statistics (Grant CLRPG3N0011) at Chang Gung Memorial Hospital and Alfred Hsing-Fen Lin and Zoe Ya-Jhu Syu for their assistance with the statistical analysis.]

Please remove any funding-related text from the manuscript and let us know how you would like to update your Funding Statement. Currently, your Funding Statement reads as follows: "This work was supported by a grant from Chang Gung Memorial Hospital, Taiwan CORPG3P0521(FCC) and CMRPG3P0801, CORPG3P0511, CORPG3P0541, CORPG3N0281, CORPG3N0282, CORPG3M0371, CORPG3M0372, CORPG3M0373, BMRPD95(SWC). This work was also supported by National Science and Technology Council grant NSTC-113-2314-B-182A-017 (FCC) and NSTC-112-2314-B-182A-107, NSTC-113-2314-B-182A-087 (SWC)."

5. We note that your Data Availability Statement is currently as follows: [All relevant data are within the manuscript and its Supporting Information files.]

Reviewers' comments:

Reviewer's Responses to Questions

**Comments to the Author**

1. Is the manuscript technically sound, and do the data support the conclusions?

Reviewer #1: Yes

Reviewer #2: Yes

Reviewer #3: Yes

Reviewer #4: Yes

2. Has the statistical analysis been performed appropriately and rigorously?

Reviewer #1: No

Reviewer #2: Yes

Reviewer #3: I Don't Know

Reviewer #4: Yes

3. Have the authors made all data underlying the findings in their manuscript fully available?

Reviewer #1: No

Reviewer #2: Yes

Reviewer #3: Yes

Reviewer #4: Yes

4. Is the manuscript presented in an intelligible fashion and written in standard English?

Reviewer #1: Yes

Reviewer #2: Yes

Reviewer #3: Yes

Reviewer #4: Yes

Reviewer #1: It is a very interesting analysis. However, do you have considered the risk indexed mortality effect during follow up period. It is not clear how you dived the quartiles, please be clear and more specific. In general all the tables should be formatted better in order to render more readable.

I think the phenomenon you observe is due tue pure increase of number, hence 'easy' cases prevalence which naturally bring to morality increase and experience of the surgeons, share of experience.

The best way is to index based of risk and complexity of the patients and see if the adjusted mortality still follows the same phenomenon.

Reviewer #2: The article does not add anything to what we already know.

It would be appropriate to review your cut-off to be considered an aortic surgeon: 5 cases per year is not enough. I have also noticed that despite some improvement over the years the mortality rate remains at around 20%, which is still quite high. How do you explain this finding? It may be related to the fact that you have included centres with very low yearly workload and high workload in the same group. It would be more appropriate to compare two separate groups: high and low workload. My understanding is that only three major centres are available. It would be worth discussing the challenges related to transfer to high volume following referral for acute and complex cases. The article would gain more value from an educational point of view if you discussed the challenging aspects and your views more extensively.

Reviewer #3: The Authors present an interesting research based on a large National administrative database.

1) unfortunately, due to inherent limitations of using administrative data, some clinical details are missing, which would be important for stratification. E.g. clinical status at presentation (shock, neurology, etc..). I understand that it's not feasible to gather such data due to the methodology of the study, which in change offers extensive follow-up information. However, this should be mentioned in the discussion section.

2) Centers are grouped by annual volume. Given the low numbers (10-20 surgeries/year), I suppose this is only referring to TAAAD surgery, not overall cardiac surgery activity. I think this should be clarified in the Methods section.

Reviewer #4: An interesting paper of Iván Alejandro De León Ayala and associates. The study is devoted to the relevant problem of the relationships between the hospital-volume and surgical outcomes of type A acute aortic dissection management. The authors showed that the operative volume inversely correlates to in-hospital mortality and postoperative complications. The volume-outcome effect extends after discharge and reflects better long-term survival. Hospital referral to high-volume centers should be considered in patients needing complex open repair for type A acute aortic syndrome. Surely, it can't be said that the authors have discovered a radical new thing. In recent years, a number of publications have noted the importance of accumulating local experience in clinics that provide care for acute aortic syndrome and the need to refer patients to such centers. These indications are present in both recently published guidelines and supplementary documents, and it is advisable to specifically note this in the text. At the same time, the work under review is important both as further evidence of the need for local aortic teams and as a study that increases the body of knowledge about predictors of complications of surgery for acute type A aortic dissection.

The authors analyzed a huge cohort of 8,059 patients in 77 hospitals who underwent first open repair for type A acute aortic dissection from 01/01/2005 to 33 31/12/2020 in Taiwan. It is important to pay attention, to indicate possible causes (e.g., DeBakey type 2 dissection of the ascending aorta), and to supplement the discussion section with the fact that not all cases of ascending aortic prosthesis were combined with intervention on the aortic arch, at least to the extent of hemiarch reconstruction, although this approach has been recommended for quite a long time. The results showed that more annual volume was significantly correlated to a lower average in-hospital mortality of the hospital (r = -0.32), but the correlation coefficient is weak.

I would also recommend double-checking the text and removing annoying typos, such as “unpreparable aortic root” (lines 261-262).

Regarding the reference list, I would recommend the authors remove reference #2. The paper entitled “The International Registry of Acute Aortic Dissection (IRAD): new insights into an old disease”, published in 2000, looks at least strange in 2025. Otherwise, the list of references is represented by contemporary and relevant works.

Generally, the paper is technically sound, and the methods are appropriate and properly conducted. The claims are fully supported by the experimental data. The statistical analysis of the data is sound. The claims are appropriately discussed in the context of previous literature. In general, the manuscript is clearly written. There are no special ethical concerns from the use of human subjects. I have no other comments on the work, and I recommend it for publication after minor revision.

Thank you for submitting your study to PLOS One and good luck with the paper.

**Do you want your identity to be public for this peer review?** For information about this choice, including consent withdrawal, please see our Privacy Policy

Reviewer #1: **Yes: ** Rafik Margaryan

Reviewer #2: No

Reviewer #3: No

Reviewer #4: **Yes: ** Vladimir Uspenskiy, MD, DrHabil

---

## [Author Response · Author response to Decision Letter 1]

12 Mar 2025

Reviewer #1: It is a very interesting analysis. However, do you have considered the risk indexed mortality effect during follow up period. It is not clear how you dived the quartiles, please be clear and more specific. In general all the tables should be formatted better in order to render more readable.

I think the phenomenon you observe is due tue pure increase of number, hence 'easy' cases prevalence which naturally bring to morality increase and experience of the surgeons, share of experience. The best way is to index based of risk and complexity of the patients and see if the adjusted mortality still follows the same phenomenon.

Response to Reviewer #1:

Thank you very much for your valuable comments. You have highlighted a very important limitation of our study: due to the data being from the National registry, we are unable to assess disease severity or risk index mortality. However, in our analysis, all of the covariates in the regression models were adjusted to age, male sex, Marfan syndrome, hypertension, Diabetes Mellitus, atrial fibrillation, chronic kidney disease, chronic obstructive pulmonary disease, heart failure, myocardial infarction, stroke, history of cardiac surgery, year of surgery, extension of aortic surgery, partial or total aortic arch replacement, elephant trunk (conventional or frozen), aortic root replacement (Bentall operation), ascending aorta replacement only, and additional surgery (Coronary artery bypass graft, valve replacement). We have discussed this issue in the revised Limitation section (Lines 303 to 307).

Thank you for the observation. We have provided a more detailed and specific description of how we divided the quartiles in our study for analysis. We have address it in the revised ‘Study population’ subsection of the Methods section (Lines 95 to 98).

Indeed, the tables were difficult to read, we have reformatted them to landscape orientation instead of portrait. Thank you for your observation.

Reviewer #2: The article does not add anything to what we already know. It would be appropriate to review your cut-off to be considered an aortic surgeon: 5 cases per year is not enough. I have also noticed that despite some improvement over the years the mortality rate remains at around 20%, which is still quite high. How do you explain this finding? It may be related to the fact that you have included centres with very low yearly workload and high

workload in the same group. It would be more appropriate to compare two separate groups: high and low workload. My understanding is that only three major centres are available. It would be worth discussing the challenges related to transfer to high volume following referral for acute and complex cases. The article would gain more value from an educational point of view if you discussed the challenging aspects and your views more extensively.

Response to Reviewer #2:

Thank you for your valuable comments. We would like to have the space to clarify a misunderstanding. In our manuscript: line 274, first sentence. We were only trying to emphasize that when a cardiovascular surgeon performs more than five aortic surgeries/year, the mortality decreases to less than 18%. We were referring to what Norton et al(1) reported in 2022. In that document, an aortic surgeon was someone performing more than 50 aortic surgeries/year.

A 20% mortality rate represents the overall mortality rate in Taiwan (High and low volume centers). Our center, which performs more than 100 aortic surgeries/year, has a mortality of less than 10%. In the report of Biancari et al(2) in 2023, although the overall mortality was 17.7%, the hospital mortality rate varied significantly between hospitals (8.9%–30.9%). This is in general, consistent with our findings.

We agree with your opinion that only three major centers are available in our country during the study period. Therefore, we conducted a sensitivity analysis of comparing patient outcomes between those who underwent surgery at the three major centers and those treated at hospitals with a very low surgical workload (fewer than 12 operations per year). The results reinforced the primary analysis, indicating better mortality outcomes for patients who underwent surgery at major centers compared to those at low-volume hospitals. We have addressed this in the revised ‘Statistical analysis’ subsection of the Methods section (Lines 138 to 140), the ‘Sensitivity analysis’ subsection of the Results section (Lines 212 to 215), and in the newly added Supplemental Table 1 and Supplemental Table 2.

Thank you very much for the advice about the transfer issue. We have added a discussion regarding referrals for acute and complex cases(3) (Lines 280 to 285).

“However, in Taiwan, there is no regionalization policy dictating the referral of patients to high-volume hospitals instead of low-volume aortic surgery centers. Thus, despite the obvious short- and long-term benefits of referring patients to high-volume centers,(3) Taiwan’s health system does not restrict where patients can undergo surgery for acute aortic dissection. It would be worthwhile for Taiwan’s health authorities to consider pilot programs or phased approaches to regionalize the surgical treatment of acute type A aortic dissection.”

References

1.Norton EL, Farhat L, Wu X, Kim KM, Fukuhara S, Patel HJ, et al. Specialization in Acute Type A Aortic Dissection Repair: The Outcomes and Challenges. Semin Thorac Cardiovasc Surg. 2023;35(3):466-75.

2.Biancari F, Juvonen T, Fiore A, Perrotti A, Herve A, Touma J, et al. Current Outcome after Surgery for Type A Aortic Dissection. Ann Surg. 2023;278(4):e885-e92.

3.Goldstone AB, Chiu P, Baiocchi M, Lingala B, Lee J, Rigdon J, et al. Interfacility Transfer of Medicare Beneficiaries With Acute Type A Aortic Dissection and Regionalization of Care in the United States. Circulation. 2019;140(15):1239-50.

Reviewer #3: The Authors present an interesting research based on a large National administrative database.

1) unfortunately, due to inherent limitations of using administrative data, some clinical details are missing, which would be important for stratification. E.g. clinical status at presentation (shock, neurology, etc..). I understand that it's not feasible to gather such data due to the methodology of the study, which in change offers extensive follow-up information. However, this should be mentioned in the discussion section.

2) Centers are grouped by annual volume. Given the low numbers (10-20 surgeries/year), I suppose this is only referring to TAAAD surgery, not overall cardiac surgery activity. I think this should be clarified in the Methods section.

1) unfortunately, due to inherent limitations of using administrative data, some clinical details are missing, which would be important for stratification. E.g. clinical status at presentation (shock, neurology, etc..). I understand that it's not feasible to gather such data due to the methodology of the study, which in change offers extensive follow-up information. However, this should be mentioned in the discussion section.

2) Centers are grouped by annual volume. Given the low numbers (10-20 surgeries/year), I suppose this is only referring to TAAAD surgery, not overall cardiac surgery activity. I think this should be clarified in the Methods section.

Response to Reviewer #3:

1) Thank you very much for the review of our manuscript. We mentioned this fact as part of our limitations. However, we have expanded the sentence to make it clearer (Lines 303 to 307):

“First, due to inherent limitations of using the Taiwan National Health Insurance (NHI) Research Database (NHIRD), we did not had access to information regarding clinical status at presentation (shock, neurology status, etc..). Disease severity was not available for study, and we did not have access to the preoperative CT scans in our study, thus, limiting stratification in our study.”

2) Thank you for your observation. We have modified the manuscript to make it clearer and avoid misunderstandings (Lines 31 to 33 within the Abstract, lines 94 to 96, and lines 148 to 149)

Reviewer #4: An interesting paper of Iván Alejandro De León Ayala and associates. The study is devoted to the relevant problem of the relationships between the hospital-volume and surgical outcomes of type A acute aortic dissection management. The authors showed that the operative volume inversely correlates to in-hospital mortality and postoperative complications. The volume-outcome effect extends after discharge and reflects better longterm survival. Hospital referral to high-volume centers should be considered in patients needing complex open repair for type A acute aortic syndrome. Surely, it can't be said that the authors have discovered a radical new thing. In recent years, a number of publications have noted the importance of accumulating local experience in clinics that provide care for acute aortic syndrome and the need to refer patients to such centers. These indications are present in both recently published guidelines and supplementary documents, and it is advisable to specifically note this in the text. At the same time, the work under review is important both as further evidence of the need for local aortic teams and as a study that increases the body of knowledge about predictors of complications of surgery for acute type A aortic dissection.

The authors analyzed a huge cohort of 8,059 patients in 77 hospitals who underwent first open repair for type A acute aortic dissection from 01/01/2005 to 33 31/12/2020 in Taiwan. It is important to pay attention, to indicate possible causes (e.g., DeBakey type 2 dissection of the ascending aorta), and to supplement the discussion section with the fact that not all cases of ascending aortic prosthesis were combined with intervention on the aortic arch, at least to the extent of hemiarch reconstruction, although this approach has been recommended for quite a long time. The results showed that more annual volume was significantly correlated to a lower average in-hospital mortality of the hospital (r = -0.32), but the correlation coefficient is weak.

I would also recommend double-checking the text and removing annoying typos, such as “unpreparable aortic root” (lines 261-262). Regarding the reference list, I would recommend the authors remove reference #2. The paper entitled “The International Registry of Acute Aortic

Dissection (IRAD): new insights into an old disease”, published in 2000, looks at least strange in 2025. Otherwise, the list of references is represented by contemporary and relevant works.

Generally, the paper is technically sound, and the methods are appropriate and properly conducted. The claims are fully supported by the experimental data. The statistical analysis of the data is sound. The claims are appropriately discussed in the context of previous literature. In general, the manuscript is clearly written. There are no special ethical concerns from the use of human subjects. I have no other comments on the work, and I recommend it for publication after minor revision.

Thank you for submitting your study to PLOS One and good luck with the paper.

Response to Reviewer #4:

Thank you very much for the appraisal of our manuscript. The typo has been corrected, and the reference #2 has been changed to a more recent publication*. We appreciate the detailed feedback.

*Biancari F, Juvonen T, Fiore A, Perrotti A, Herve A, Touma J, et al. Current Outcome after Surgery for Type A Aortic Dissection. Ann Surg. 2023;278(4):e885-e92.

---

## [Decision Letter · Decision Letter 1]

31 Mar 2025

Dear Dr. Chen,

Thank you for submitting your manuscript to PLOS ONE. After careful consideration, we feel that it has merit but does not fully meet PLOS ONE’s publication criteria as it currently stands. Therefore, we invite you to submit a revised version of the manuscript that addresses the points raised during the review process.

**ACADEMIC EDITOR: ** The authors are thanked for this submission to the PLOS One. However, after a critical external peer review by three experts in the field, I recommended that you incorporate recent literature findings and address the concerns raised by peer reviewers to enhance the clarity of the discussion section in your paper.

Please see the attached reviewer comments detail below.

https://journals.plos.org/plosone/s/submission-guidelines#loc-laboratory-protocols . Additionally, PLOS ONE offers an option for publishing peer-reviewed Lab Protocol articles, which describe protocols hosted on protocols.io. Read more information on sharing protocols at https://plos.org/protocols?utm_medium=editorial-email&utm_source=authorletters&utm_campaign=protocols .

We look forward to receiving your revised manuscript.

Kind regards,

Dr Redoy Ranjan, MBBS, MRCSEd, Ch.M., MS (CV&TS), FACS

Academic Editor

PLOS ONE

Journal Requirements:

Reviewers' comments:

Reviewer's Responses to Questions

**Comments to the Author**

Reviewer #1: All comments have been addressed

Reviewer #2: All comments have been addressed

Reviewer #3: All comments have been addressed

2. Is the manuscript technically sound, and do the data support the conclusions?

Reviewer #1: Yes

Reviewer #2: Yes

Reviewer #3: Yes

3. Has the statistical analysis been performed appropriately and rigorously?

Reviewer #1: Yes

Reviewer #2: Yes

Reviewer #3: Yes

4. Have the authors made all data underlying the findings in their manuscript fully available?

Reviewer #1: No

Reviewer #2: No

Reviewer #3: Yes

5. Is the manuscript presented in an intelligible fashion and written in standard English?

Reviewer #1: Yes

Reviewer #2: Yes

Reviewer #3: Yes

Reviewer #1: Dear authors congratulations for topic.

Massive transfusion is defined as > 10 U, define also the single unit of transfused blood.

Are there any frozen elephant trunk cases? Quartile's division cutoff values must be also in methods section. In Figure 1 C could be helpful to have linear regression line. From figure 2 is visible that hospitals with less then 5 cases year have highest mortality, visually over 50%, so I suggest to create a version of the same plot excluding hospitals with less then 5 cases, or at least to do separate plots.

In figure 3 there is flipped phenomenon of cumulative mortality, it seems third quartile is doing better then forth one, why do you think?

In figure 4D there is an increasing rend after around 27 cases, how you explain it?

Reviewer #2: (No Response)

Reviewer #3: The Authors have addressed all the issues raised by the Reviewers, improving the paper throughout, therefore the article is now acceptable in my opinion.

**Do you want your identity to be public for this peer review?** For information about this choice, including consent withdrawal, please see our Privacy Policy

Reviewer #1: **Yes: ** Rafik Margaryan

Reviewer #2: No

Reviewer #3: No

---

## [Author Response · Author response to Decision Letter 2]

24 Apr 2025

Reviewer #1: Dear authors congratulations for topic.

1. Massive transfusion is defined as > 10 U, define also the single unit of transfused blood.

Thank you for the observation. A single unit equals to 450-500cc of blood. We have added this definition in line 106.

2. Are there any frozen elephant trunk cases?

As shown in Table 1, there were 1,269 (15.7%) cases of frozen elephant trunk (Few were conventional elephant trunk).

3. Quartile's division cutoff values must be also in methods section.

Thank you very much for observation. We have added the quartile's division cutoff values into the revised ‘Study population’ subsection of the Methods section. (Lines 97-98)

4. In Figure 1 C could be helpful to have linear regression line.

Thank you very much for pointing this out. We have added the linear regression line to revised Figure 1C.

5. From figure 2 is visible that hospitals with less then 5 cases year have highest mortality, visually over 50%, so I suggest to create a version of the same plot excluding hospitals with less then 5 cases, or at least to do separate plots.

Thank you for the suggestion. We have created a separate plot excluding hospitals with less than 5 cases per year. We have addressed this in the newly added Supplemental Figure and the revised ‘Sensitivity analysis’ subsection of the Results section. (Lines 213-215)

6. In figure 3 there is flipped phenomenon of cumulative mortality, it seems third quartile is doing better than the fourth one, why do you think?

This is an interesting observation. One plausible explanation is that within the highest-volume centers (4th quartile), some centers handle a high volume of aortic dissections specifically but are not necessarily high-volume centers for general open-heart surgery. Consequently, their overall surgical experience and infrastructure may differ, potentially impacting outcomes negatively compared to centers proficient in both high-volume open-heart surgery and acute type A dissections. This variability could artificially elevate cumulative mortality within the 4th quartile, making outcomes appear worse than the 3rd quartile. Supporting this interpretation, Figure 3B, which excludes the two Highest-Volume Centers, shows improved cumulative mortality in the 4th quartile group, indicating that these specific high-volume but less comprehensive centers may skew the overall outcomes negatively. We have described this in the revised ‘Sensitivity analysis’ subsection of the Results section. (Lines 223-229)

7. In figure 4D there is an increasing rend after around 27 cases, how you explain it?

In Figure 4D, an increasing trend after approximately 27 cases can be explained by the practice patterns of high-volume centers. Such institutions typically have more rigorous postoperative surveillance protocols, including frequent imaging studies (e.g., CT scans, echocardiograms). Consequently, they are more likely to detect indications for early re-intervention. Additionally, high-volume centers possess greater surgical experience and expertise, which makes them more proactive in scheduling and performing reoperations compared to lower-volume centers.

Reviewer #2: (No Response)

Response: Thank you very much for review.

Reviewer #3: The Authors have addressed all the issues raised by the Reviewers, improving the paper throughout, therefore the article is now acceptable in my opinion.

Response: Thank you very much for review.

---

## [Decision Letter · Decision Letter 2]

8 May 2025

Dear Dr. Chen,

Thank you for submitting your manuscript to PLOS ONE. After careful consideration, we feel that it has merit but does not fully meet PLOS ONE’s publication criteria as it currently stands. Therefore, we invite you to submit a revised version of the manuscript that addresses the points raised during the review process.

**ACADEMIC EDITOR:** While we found the work interesting, it would benefit from minor revision. Therefore, I invite you to respond to the reviewer's comments and submit a revised version of your article.

We look forward to receiving your revised manuscript.

Kind regards,

Redoy Ranjan, MBBS, MRCSEd, Ch.M., MS (CV&TS), FACS

Academic Editor

PLOS ONE

Journal Requirements:

Reviewers' comments:

Reviewer's Responses to Questions

**Comments to the Author**

Reviewer #1: All comments have been addressed

2. Is the manuscript technically sound, and do the data support the conclusions?

Reviewer #1: No

3. Has the statistical analysis been performed appropriately and rigorously?

Reviewer #1: Yes

4. Have the authors made all data underlying the findings in their manuscript fully available?

Reviewer #1: No

5. Is the manuscript presented in an intelligible fashion and written in standard English?

Reviewer #1: Yes

Reviewer #1: Please do share your data according to PLOS data policy requirements.

The patients characteristics should be in methods section. It seems that third quartile is doing better then fourth one in immediate results (table 2).Please do add a discussion on this topic. What is a meaning of table 5 excluding second and third most volume centers?

**Do you want your identity to be public for this peer review?** For information about this choice, including consent withdrawal, please see our Privacy Policy

Reviewer #1: **Yes: ** Rafik Margaryan

---

## [Author Response · Author response to Decision Letter 3]

14 May 2025

Reviewer #1: Please do share your data according to PLOS data policy requirements. The patients characteristics should be in methods section. It seems that third quartile is doing better then fourth one in immediate results (table 2). Please do add a discussion on this topic. What is a meaning of table 5 excluding second and third most volume centers?

1. Response: Thank you very much for your thoughtful review. We would like to respectfully note that the issues raised have already been addressed in our previous revisions. Specifically, the points mentioned were incorporated based on earlier reviewer comments, and corresponding changes have been made in the manuscript.

---

## [Editor Report · Decision Letter 3]

19 May 2025

Hospital Volume and Outcomes of Surgical Repair in Type A Acute Aortic Dissection: A Nationwide Cohort Study

PONE-D-25-00813R3

Dear Dr. Chen,

We’re pleased to inform you that your manuscript has been judged scientifically suitable for publication and will be formally accepted for publication once it meets all outstanding technical requirements.

Kind regards,

Dr Redoy Ranjan, MBBS, MRCSEd, Ch.M., MS (CV&TS), FACS

Academic Editor

PLOS ONE
---

## [Editor Report · Acceptance letter]

PONE-D-25-00813R3

PLOS ONE

Dear Dr. Chen,

I'm pleased to inform you that your manuscript has been deemed suitable for publication in PLOS ONE. Congratulations! Your manuscript is now being handed over to our production team.

Kind regards,

on behalf of

Dr. Redoy Ranjan

Academic Editor

PLOS ONE